# Effect of Celeriac Pulp Maceration by *Rhizopus* sp. Pectinase on Juice Quality

**DOI:** 10.3390/molecules27238610

**Published:** 2022-12-06

**Authors:** Grażyna Jaworska, Natalia Szarek, Paweł Hanus

**Affiliations:** 1Dept. Food Technology and Human Nutrition, Institute of Food and Nutrition Technology, College of Life Sciences, University of Rzeszow, St. Zelwerowicza 4, 35-601 Rzeszów, Poland; 2Doctoral School of the University of Rzeszów, University of Rzeszów, St. Rejtana 16C, 35-959 Rzeszów, Poland

**Keywords:** celeriac juice, pectinase from *Rhizopus* sp., antioxidant activity, phenolic compounds, physicochemical studies

## Abstract

Peeled and unpeeled celeriac pulp was macerated with pectinase from *Rhizopus* sp. at 25 °C for 30 and 60 min. Peeling, enzyme addition, and maceration time significantly affected the quality characteristics of the juice. The juice obtained from peeled celeriac was characterized by higher pressing yield, sucrose content, and antioxidant activity (ABTS^*+^ and DPPH^*^). The juice from the unpeeled root had higher extract, fructose, glucose, total polyphenols, antioxidant activity (FRAP), total phenolic acids, and total luteolin content. Applying the enzyme to celeriac pulp had no significant effect on the extract’s content, analyzed sugars, and antioxidant activity of the juices (ABTS^*+^). Adding pectinase to unpeeled celery pulp resulted in a 2–10% increase in pressing efficiency, compared to the control sample held at 25 °C for the same period. Maceration of the enzyme-peeled pulp increased the antioxidant potential of the juice by 22% in the FRAP method. In contrast, in all juices analyzed, unpeeled and peeled roots increased antioxidant activity measured by the DPPH* method by 24–57% and total phenolic acids by 20–57%. The time of holding the pulp at 25 °C was an important factor, and its extension resulted in a decrease in the values of most of the analyzed parameters, with the exception of pressing efficiency and fructose content in all analyzed juice samples. Short-term, 30-min maceration of peeled and unpeeled celery pulp with pectinase from *Rhizopus* sp. had a significant effect on increasing juice yield, antioxidant activity, and phenolic compound content.

## 1. Introduction

Celeriac (*Apium graveolens* L. var. *rapaceum*) is a biennial plant of the *Apiaceae* family, native to temperate climates [1]. It is a low-calorie vegetable (about 40 kcal/100 g FM) with a low basic nutrient content [1]. In 100 g of FM, there are 0.8–1.2 g of protein, 0.2–0.3 g of fat, and 3.2–9.7 g of carbohydrates [2]. Celeriac is also rich in fiber (approximately 2.0 g/100 g FM.) and micro and macro elements (Fe, Cu, Zn, Mn, B, Ca, K, Mg, P, S). Significant amounts of potassium and sodium are responsible for fluid regulation in the body, preventing the development of urinary tract diseases [3]. This vegetable can also be eaten to prevent complications caused by diabetes. Celeriac stimulates the pancreas to secrete insulin, reducing glucose and lipids in the blood [4,5]. In addition, it has a beneficial effect on the immune system and pharmacodynamic activity [6].

This vegetable’s distinctive and characteristic taste and smell result from aldehydes, esters, terpenes, low molecular sugars, organic acids, and flavonoids, especially phthalides such as sedanolide and 3-n-butylphthalide, in the celeriac composition [7]. The latter compounds exhibit anticarcinogenic and neuroprotective properties and reduce blood pressure by expanding the smooth muscles in the blood vessels [8]. The antioxidant activity of celeriac is due to the presence of polyphenols, which are responsible, i.e., for the free radicals scavenging, the ability to absorb single oxygen, or the transition metal chelation [9,10,11]. The key role of apigenin and luteolin, the main flavonoids of celeriac, is to prevent and inhibit cardiovascular inflammation, thrombosis, and hyperlipidemia, regulate the work of the heart muscle and reduce the risk of osteoporosis by protecting the skeletal system against weight loss [12]. Furthermore, polyphenols play an important role in lipid peroxidation and inhibition of various oxidative enzymes [13,14,15]. Celeriac also shows hepatoprotective, antibacterial, antiviral, antifungal, and analgesic effects [1,3,4].

There is little literature on innovative methods of processing vegetable pulp or on the technology of pressing vegetable juices, especially from celery. Studies are mainly focused on the enzymatic treatment of fruits. Concerning vegetables, most research refers to the methods of juice production from macerated carrot tissue. According to Nowak [16], to inactivate native enzymes and remove microorganisms and spore forms of bacteria, the vegetable pulp should be heated at 110–120 °C. After cooling to 45–50 °C, the enzymatic liquefaction is carried out at pH 4.5 for 60 min. However, the use of elevated temperatures contributes to the degradation of health-promoting components [17].

Pectinases, the enzymes responsible for pectin degradation, are found in the middle lamina and the primary cell walls of plants and contribute to juice clouding [18,19]. Pectinases of microbiological origin account for about 1⁄4 of the global production of food and industrial enzymes [20]. These enzymes have been used in the food industry to ferment tea and coffee leaves, extract essential oils and produce new functional products [21,22,23]. They are particularly important for clarifying juices and wines [16]. Enzymatic degradation of pectins into smaller molecules after gentle mechanical treatment significantly improves the properties of the end product. When used in the processing of fruit and vegetable pulp, it contributes to the viscosity of the reduction of the juice and to the removal of opacities, thus supporting the process of clarification, filtration, and stabilization of the end products [19,24,25]. Pectinolytic enzymes were already used to produce fruit juices in the 1930s. They were used most often to depectinize some fruit juice and to process pulp and berry juice [26]. A properly selected enzyme preparation or enzyme supports the pressing process, prevents the clouding of juices caused by an excessive amount of hemicellulose, and facilitates clarification and filtration [16]. Pectinase treatment of apple and berry juices, as well as guava puree, significantly improved antioxidant capacity, total phenolic content, increased process efficiency, and sugar concentration, making the color of the end products more attractive [27]. For vegetable tissue processing, liquid pectinolytic preparations are most often used. These are Pectinex Ultra SP-L, Ultrazym AFP-L, Rapidase^®^ Carrot, Rohament PL^®^, Panzym SMASH XXL^®^ or ROHAPECT^®^ MA Plus, as well as powdered *Aspergillus* sp. pectinase [25].

The aim of this study was to assess the effect of enzymatic maceration of celeriac pulp, obtained from peeled and unpeeled celeriac, on the juice quality. The process was performed by *Rhizopus* sp. pectinase at its optimal temperature (25 °C). The selected physicochemical parameters, the content of selected carbohydrates, antioxidant activity, and the phenolic compounds profile of the juices obtained were examined.

## 2. Results and Discussion

*Aspergillus* sp. pectinase has the highest activity at pH 4.0 and 50 °C. Due to the relatively high temperature of the process and possible degradation of bioactive compounds, in this work, pectinase was used, with optimum activity at a temperature slightly exceeding room temperature. *Rhizopus* sp. pectinase is used for enzymatic digestion of the cell wall. It is not only a source of pectinase activity but also shows cellulase and hemicellulose activity. It shows the highest enzymatic activity at pH 4.0 and a temperature of 25 °C. Thus, it can be assumed that the enzyme treatment will allow more thermolabile and health-promoting components to be retained in the end products.

### 2.1. Pressing Yield, pH, Content of Total Soluble Solids and Selected Sugars in the Celeriac Juice

The results of pressing efficiency, pH, extract content, and selected carbohydrates in the juices are presented in Table 1. The pressing efficiency of the juices obtained from peeled celeriac fluctuated between 47% and 67% and was the highest in the control sample, kept at 25 °C for 60 min (PC2). Higher efficiencies (49–70%) were achieved after pressing juices from the unpeeled celeriac pulp, while the highest was observed in samples having undergone one-hour enzymatic maceration (UCE2). The longer incubation time of the pulp at 25 °C resulted in a higher pressing efficiency in all samples examined. Nadulski et al. [1] found that the pressing yield of the juice from celeriac pulp was about 54%, which was close to the pressing yield of the enzyme untreated and not incubated control (PC and UC). Nowak and Tempczyk [28] showed that using enzyme preparations improves the pressing efficiency by an average of 16–22% compared to the control.

Our study showed that using pectinase in the pulp of unpeeled celeriac significantly increased the juice yield, but an average of only 2 to 10% compared to the control. In turn, an adverse effect of enzyme use was observed in the juice obtained from peeled celeriac, which had an average yield 14–17% lower than the control samples kept at 25 °C for 60 min (PCE2) and 30 min (PCE1). The more beneficial effect of enzymatic maceration of the pulp from unpeeled celeriac on the juice yield may be due to better digestion of the hard celeriac tissue, resulting in its greater loosening and, consequently, higher efficiency compared to the juice obtained from the peeled celeriac.

The pH value of the juice from peeled and unpeeled celeriac varied between 5.8 and 6.0. Profir and Vizireanu [12] also showed similar pH values in the examined celeriac. According to the assumption made in this study, the pH of the pulp treated with pectinase was within the range of 4.0 (3.9 for UCE1 and UCE2 and 4.2 for PCE1 and PCE2). Extending the pulp incubation time to 25 °C did not have a significant effect on the parameter examined.

The average content of extract in celeriac juices ranged from 6.7 to 8.3 °Bx. In juices from unpeeled celeriac, their content was significantly higher (8.0–8.3 °Bx) than those from peeled celeriac (6.7–6.9 °Bx). Incubation of the pulp of the peeled root with pectinase increased the extract by 0.5 °Bx (PCE1 and PCE2), while 60-min maceration of unpeeled celeriac (UCE2) pulp reduced the extract to 7.7 °Bx. The extract content, reported by Nadulski et al. [1] in celeriac juices, was higher than in our study and ranged from 8.4 to 8.9 °Bx. The difference in this parameter between our studies and the findings of other authors may result from using various cultivars as the raw material and from the differences in the substance content in dry matter of the raw material [29].

The juices of the unpeeled celeriac had significantly more fructose and glucose, while sucrose was the predominant carbohydrate in the juices pressed of the peeled celeriac. Concerning sugar contents, 1 L of celeriac juices contained: 2.8–6.3 g of fructose, 14.9–25.0 g of glucose, and 15.7–22.1 g of sucrose. The highest sucrose content in the juice was observed after 60-min incubation of peeled celeriac (PC2) pulp at 25 °C and after maceration of pulp with pectinase (PCE2), and these values did not differ significantly (22.0–22.1 g/L). The lowest amount of this disaccharide was found in the juices obtained from unpeeled celeriac, kept at 25 °C for 60 min (UC2—15.7 g/L). The application of pectinase to pulp had no significant effect on the sucrose content in peeled celeriac juice and the fructose and glucose content in unpeeled celeriac juice. However, maceration of the pulp with the enzyme reduced the level of fructose and glucose in the juice of peeled root, and the 60-min enzyme activity in the pulp from unpeeled celeriac contributed to the increase in sucrose in the juice, which may indicate the ongoing process of complex carbohydrate hydrolysis (enzymatic or acidic).

### 2.2. Determination of the Antioxidant Activity in the Celeriac Juices

Table 2 shows the values of the celeriac juices’ antioxidant potential and total polyphenol content. The antioxidant activity of the juices, measured by the ABTS^*+^ assay, ranged from 3666 to 4759 μmol Trolox/L. The highest values were recorded in juices pressed from peeled (PC) and unpeeled (UC) celeriac, without enzyme and not incubated (4759 and 4638 μmol Trolox, respectively) as well as in those obtained after 1 h of maceration of the peeled celeriac pulp with pectinase (PCE2–4626 μmol Trolox/L). The longer incubation time of unpeeled celeriac pulp, including those incubated by pectinase, reduced the antioxidant potential, whereas this relationship was reversed in the analysis of juices from PC. According to Yao and Ren [14], who measured the antioxidant activity using the ABTS^*+^ assay, the antioxidant activity of celeriac varieties Tropica and Shengije were approximately 585 μM and 835 μM Trolox/100 g FW, respectively. Different values of activity, reported by Yao and Ren [14], could have been caused by using a different celeriac variety or a different sample preparation for analysis, i.e., the prior blanching of the raw material, drying, and deep freezing of the research material.

The DPPH^*^ radical scavenging activity in celeriac juices ranged from 449 to 1108 µmol Trolox/L and was the highest in juices from the pectinase-macerated celeriac pulp. The enzymatic maceration of celeriac pulp peeled for 30 (PCE1) and 60 min (PCE2) contributed to an increase in the antioxidant capacity of the juices by 57% and 31%, respectively, compared to the control samples kept at the same time at 25 °C. In the juice obtained from the pulp of UCE1, radical activity increased by approximately 41% compared to the same control sample. Yao and Ren [14] showed that the radical activity measured by the DPPH^*^ assay in the extracts of varieties Tropica and Shengije was 612 and 734 μM Trolox/100 g FM, respectively.

The antioxidant power reducing ferric ions, measured by the FRAP assay, ranged from 96 to 401 μmol Trolox/L and was highest in UC juice (unpeeled celeriac, no enzyme, zero incubation time) and about 4 times higher than in PC juice (PC). Keeping unpeeled celeriac pulp with pectinase for 30 min (UCE1) significantly reduced the antioxidant activity by approx. 52% compared to the control sample (UC1). The prolongation of the enzyme incubation time did not significantly affect the value of the examined parameter. The application of pectinase to the PC pulp significantly increased the antioxidant activity of the analyzed juices by about 22% compared to the same control sample.

The total polyphenol content fluctuated between 93 mg and 225 mg GAE/L. The addition of pectinase and the incubation time had a statistically significant negative effect on the total polyphenol content of the juices examined. The lowest amount of polyphenolic compounds was found in UCE2 juice, and the highest was in the control sample pressed from the UC pulp (no enzyme, zero incubation time). Salamatullah et al. [30] revealed that after heat treatment, the polyphenol content in 100 g celeriac decreased from 22.2 mg GAE in control to 3.0 mg GAE. In turn, the total polyphenol content recorded by Yao and Ren [14] in the celeriac cultivar Tropica was 26 mg GAE/100 g FM, while in the cultivar Shengije this value was higher and amounted to 35 mg GAE/100 g FM.

### 2.3. Identification of Phenolic Compounds in the Celeriac Juice

Table 3 presents the compounds identified in celeriac juice based on publications by various authors, along with characteristic retention times (tr), maximum absorbance wavelength (Λ_max_), deprotonated molecules ([M-H]-MS/MS), and major fragmentation ions. In all celeriac juices, a total of 6 phenolic acids and 14 flavonoids were identified. Each phenolic compound listed in Table 3 was identified in all juices analyzed pressed from both peeled and unpeeled celery. Apart from ferulic acid (*m/z* 193), quinic acid (*m/z* 191), and dicaffeic acid (*m/z* 341), there were also caffeic acid hexoside (*m/z* 341), coumaroylquinic acid (*m/z* 337), and caffeoylsinpylquinic acid (*m/z* 559). Celeriac juices also contained chrysoeriol (*m/z* 299), chrysoeriol-7-O-6′′-malonyl glucoside (*m/z* 547), chrysoeriol-7-O-apiosylglucoside (*m/z* 593), and chrysoeriol-7-O-glucoside (*m/z* 461). Among luteolins, the following were identified: luteolin-7-O-malonyl-apiosylglucoside B (*m/z* 665); luteolin-7-O-glucoside (*m/z* 447); acetylated luteolin hexoxyl-rhamnoside (*m/z* 635); luteolin-7-O-malonyl-apiosylglucoside B (*m/z* 635). The juices also included apigenins such as apigenin-6-C-glucoside (*m/z* 431), apiin (*m/z* 563), and acetylated apigenin-C-hexoside-O-pentoside (*m/z* 605). Furthermore, hydroxydimethoxycoumarin (*m/z* 221), kaempferol-3,7-O-diglucoside (*m/z* 611), and taxifolin hexoside I and II (*m/z* 465) were also identified in such juices.

### 2.4. Content of Selected Phenolic Acids, Apigenins, and Luteolins in the Celeriac Juice

The content of phenolic acids, apigenin, and luteolin is shown in Table 4. Of the total content of phenolic acids, the juice examined had 74.9–98.1% quinic acid, 0.1–10.2% dicaffeic acid, and 1.8–23.1% ferulic acid. A significantly higher content of phenolic acids was found in the UC juices. The pectinase addition to celeriac pulp substantially increased the content of analyzed acids in juices. Their highest amounts were found in PCE1 and UCE1 juices. It was also observed that the extension of the enzymatic maceration time of pulp had a detrimental effect on the total phenolic acid content of phenolic acids in the analyzed juices. Ertekin Filiz et al. [31] also determined selected phenolic acids in vegetable broths and fresh vegetables, including celeriac, which contained gallic acid (11–57 mg/L), chlorogenic acid (25–64 mg/L), caffeic acid (10–86 mg/L), p-coumaric acid (10–82 mg/L and ferulic acid (13–82 mg/L). This means that the total content of phenolic acids in fresh celeriac was between 69 and 371 mg/L. Therefore, these values correspond to the results obtained for the examined celeriac juices.

Guerra et al. [32] and Fazal and Singla [33] claim that the main antioxidants of celeriac are luteoline and apigenin and the content of these flavonoids is between 3.70 and 8.14 mg QA per 100 g fresh weight. In vitro apigenin shows anti-aggregation activity and inhibits contractions of isolated smooth muscles of the thoracic aorta [6]. In addition, it reduces free radicals and promotes metal chelation. Luteolin, on the other hand, can prevent DNA changes caused by carcinogens, lowers cholesterol and diabetes, and prevents cardiovascular and neurodegenerative diseases [34]. Total apigenin content was expressed as the sum of apigenin-6-C-glucoside, apigenin, and acetylated apigenin-C-hexoside-O-pentoside. Their amount in celeriac juices was respectively 4.8–60.0%, 20.0–90.2%, and 3.4–20.0% of the total apigenin content. Significantly higher values of total apigenin content were recorded in UC juices. The highest total apigenin content showed the UC2 juice (1 h Incubation period, 25 °C). The application of pectinase to PC pulp had no significant effect on the apigenin content of celeriac juices. In contrast, the apigenin content of the juice from UC pulp juice decreased by 8–33% compared to control samples kept at 25 °C for the same period.

The luteolin content was expressed as the sum of the three flavonoids. In celeriac juice, the following were determined: luteolin-7-O-glucoside (50.0–87.1% of the total sum of luteolin), acetylated luteolin-hexoxyl-rhamnosyde (6.5–33.3%), and luteolin-7-0-malonyl-apiosylglucoside (0.0–30.0%). Luteolin content was significantly higher in unpeeled and enzyme-free control samples (UC, UC1, and UC2). There was no significant effect of maceration of the PC pulp on the luteolin content in the juices; however, a significant adverse effect on its content was observed in the juices from the UC pulp (a reduction of 58–81% compared to the control). A longer incubation period at 25 °C had a detrimental effect on all juices examined, except for UCE2 juice from (unpeeled celeriac, 1 h pectinase maceration). Citing Dietrich et al. [35], Gheribi [36] states that celeriac contains approximately 4.6 mg/100 g of apigenin and 1.3 mg/100 g of luteolin.

### 2.5. Colour Assessment in the Celeriac Juices by the CIE L*a*b* System

Table 5 presents the colour parameters that determine the lightness of the juices (L*), the green/red proportion (a*), and the yellow/blue proportion (b*). All examined color parameters differed statistically significantly between the examined samples tested. The values of the L* parameter ranged from 31.87 to 36.95. Juices pressed from unpeeled, enzyme-free, and not incubated celeriac pulp (UC-L* = 36.95) were the lightest, while the darkest were those from unpeeled and peeled celeriac pulp, macerated with pectinase. Negative values for parameter a*, indicating a higher proportion of green color, were observed in both UC and PC juices, including those macerated with pectinase. The highest proportion of red color, as indicated by the positive values of this parameter, was determined only in unpeeled celeriac juices without enzyme, and ranged from 0.63 in the pulp juice incubated for 1 h at 25 °C (UC2) to 2.22 in fresh pulp juice, not incubated at 25 °C (UC). The positive values of the b* parameter, determining the share of yellow colour, ranged from 2.82 to 9.54 and were higher in UC juices. The application of pectinase to celeriac pulp significantly reduced the parameter b*, indicating a less intense yellow colour of the juices examined compared to the control. The total color difference for PC juices was 0.60–0.76 for celeriac pulp kept at 25 °C for 30 and 60 min and 2.11–2.38 for pectinase-macerated pulp. The largest visible color differences were observed in UC juices after pectinase treatment (UCE1-ΔE = 7.45 and UCE2-ΔE = 7.81) compared to those obtained from the pulp without enzyme, but after 30 and 60-min incubation at 25 °C (UC2-ΔE = 3.40 and UC1-ΔE = 4.39). Our studies showed that the maceration of the pulp with pectinase significantly affected the investigated juices’ color, especially the parameter b*.

The parameters a* and b* were clo”ely ’orrelated with the content of total polyphenols, apigenins, and luteolins (r = 0.74–0.92 and r = 0.78–0.87, respectively). Furthermore, the parameter L* was strongly and positively correlated with the total polyphenol content (r = 0.73).

## 3. Materials and Methods

### 3.1. Chemicals and Reagents

TPTZ reagent was purchased from Fluka Analytical (Buchs, Switzerland), formic acid from Fisher Scientific (Waltham, MA, USA), DPPH^*^ and ABTS^*+^ reagents, as well as pectinase enzyme from *Rhizopus* sp. were supplied by Sigma-Aldrich Company (Saint Louis, MO, USA). Acetonitrile, methanol, Folin-Ciocalteau reagent, and acetate buffer (pH 3.6) were purchased from Chempur (Piekary Śląskie, Poland), and sodium carbonate from POCH (Gliwice, Poland).

### 3.2. Plant Materials

The experimental material was the celeriac of the *Apium graveolens* L var. Zagłoba, purchased from a warehouse (Rzeszów, Poland) at the turn of January/February 2022. The roots were characterized by optimal functional and organoleptic properties and did not show physical or microbiological damage. They were processed immediately after purchase.

### 3.3. Processing

The celeriacs were sorted, washed under running water, and divided into two parts: peeled and unpeeled. The raw material was manually cut into 3 cm × 3 cm pieces and pulped with a Vorwerk-Thermomix TM31 multifunctional machine (Wuppertal, Germany). The pulp obtained was brought to 25 °C, and then 300 g was weighed for each of the 10 trials. The samples intended for incubation with pectinase were previously acidified with 80% citric acid at pH 4.0 and then macerated with the enzyme (30 mg/300 g of pulp) at 25 °C, a temperature optimal for *Rhizopus* sp. pectinase, for 30 and 60 min. Control samples, prepared from peeled and unpeeled celeriac without the enzyme addition, were kept at 25 °C for 30 and 60 min. At the end of the maceration period, the juice was pressed out of the pulp using a Norwalk-Juicers type 275 hydraulic press (Seattle, WA, USA). The experiment was carried out in three independent replicates.

### 3.4. Determination of the Pressing Yield, pH, Content of Total Soluble Solids, and Selected Sugars

Pressing yield was determined by the weight-volume method according to the formula: Wj = [M/Mp]*100%, where (pressing yield [%], M—the weight of juice [g], and Mp—the initial weight of pulp [g]).

The pH measurement was performed at 20 °C ± 1 °C using the pH meter CP-401 (Elmetron, Zabrze, Poland), according to PN-EN 1132:1999 [37].

The total soluble solids (°Bx) were measured refractometric at 20 ± 1 °C with an Abbe Zuzi type 325 refractometer with a Brix scale, according to PN-EN 12143:2000 [38].

The contents of fructose, glucose, and sucrose were determined using high-performance liquid chromatography (HPLC). Juice samples intended for analysis were previously centrifuged in a laboratory centrifuge MPW-260R (Warsaw, Poland) at 7300 rpm/min for 10 min and filtered through PTFE syringe filters with a pore size of 0.45 µm. The content of selected sugars (fructose, glucose, and sucrose) was determined using a high-performance HPLC liquid chromatograph (SYKAM, Fürstenfeldbruck, Germany), controlled by the Clarity Software version 6.1 program (UK), and equipped with the Sykam S1132 and S1130 high-pressure pumps, the Sykam S5300 sample injector, and the Sykam S3585 refractometer detector. Separation was carried out on a Cosmosil SUGAR-D column (4.6 mm × 250 mm). Elution was carried out at a flow rate of 1 mL/min, maximum pressure of 100 Ba, and a temperature of 35 ± 5 °C. The mobile phase consisted of acetonitrile (phase A: 75%) and water (phase B: 25%). The injection volume was 20 µL and the total analysis time was 15 min. The quantitative analysis was performed based on a standard curve for fructose, glucose, and sucrose as reference substances. Analysis was carried out in triplicate for each juice sample, from which the arithmetic mean (n = 9) was taken. The results are given in g per 1 L of juice.

### 3.5. Determination of the Antioxidant Activity

The juice samples to be analyzed were previously centrifuged at 7300 rpm/min for 10 min in a laboratory centrifuge MPW-260R (Warsaw, Poland). Afterward, methanol extracts from the decanted samples were prepared and appropriately diluted, depending on the analysis. Three replicates were prepared for each sample. Methanol extracts were used to measure the antioxidant activity of celeriac juices using the ABTS^*+^, DPPH^*^, and FRAP methods and to determine total polyphenol content by the Folin-Ciocalteau method.

#### 3.5.1. Determination of Antioxidant Activity Using ABTS^*+^ Cation Radical

Determination of antioxidant activity with ABTS^*+^ radical cation was performed according to Re et al. [39], and the decrease of absorbance at the wavelength λ = 734 reflected antioxidant concentration. The absorbance was measured against distilled water, 6 min after the initiation of the reaction, using a spectrophotometer UV-1900 UV-Vis Shimadzu (Kioto, Japan). The final results are expressed as μmol Trolox/L juice.

#### 3.5.2. Determination of Antioxidant Activity Using the DPPH^*+^ Method

Antioxidant activity against the DPPH^*^ (1,1-diphenyl-2-picrylhydrazyl) radical was determined using a method developed by Yen and Chen [40]. The absorbance was measured against 96% methanol after 10 min of reaction at a wavelength of λ = 517 nm. The absorbance was measured using the same spectrophotometer in Section 3.5.1. The final results are expressed as µmol Trolox/L of juice.

#### 3.5.3. Measurement of Ferric Ion–Reducing Antioxidant Power by the FRAP Assay

The ferric [Fe(III)] ion–reducing antioxidant power (FRAP) assay was performed according to Benz et al. [41]. Measurement was carried out 10 min from the initiation of the reaction at the wavelength λ = 595 nm. The absorbance was measured using the same spectrophotometer in Section 3.5.1. The antioxidant activity is given as μmol Trolox/L juice.

#### 3.5.4. Determination of Total Polyphenols Content

The total polyphenol content (TPC) was determined by the Folin-Ciocalteau method, using the method developed by Xianggun et al. [42]. The absorbance was measured using the same spectrophotometer in Section 3.5.1 at 765 nm, 60 min after the initiation of the reaction, against distilled water. Over this time, the sample was kept in a shaded place. The results are presented as mg of gallic acid equivalent (GAE) per 1 L of celeriac juice.

#### 3.5.5. Identification and Quantification of Phenolic Compounds

Immediately before the analysis, the juice samples were centrifuged for 10 min at 7300 rpm/min in an MPW-260R laboratory centrifuge (Warsaw, Poland) and filtered through a 0.45 μm PTFE syringe filter. Analysis was performed using an Acquity UltraPerformance Liquid Chromatograph (UPLC-PDA-MS/MS; Waters, Micromass, Manchester, UK), coupled with a photodiode matrix detector (PDA) and a tandem quadrupole mass detector (TQD) equipped with an electrospray ionization (ESI) source. Separation of the phenolic compounds was performed on a UPLC BEH RP C18 column (1.7 μm, 100 mm × 2.1 mm i.d., Waters Corporation, Milford, MA, USA). The mobile phase was 0.1% aqueous formic acid solution (solvent A-90%) and 40% acetonitrile dissolved in water (solvent B-10%). The elution was performed as follows: start at 0 min, 5% B; 0–8 min, linear gradient up to 100% B; 8–9.5 min for washing and return to initial conditions. The flow rate was kept constant at 0.3 mL/min throughout the run of the analysis period (13 min). The injection volume of the samples was 5 μL, and the column temperature was maintained at 50 °C. The following TQD parameters were used: cone voltage 30 V; capillary voltage 3500 V; source and desolvation temperature: 120 °C and 350 °C, respectively; and a desolvation gas (argon) flow rate of 800 L/h. The individual phenolic compounds were characterized based on the retention time, mass-to-charge ratio, fragmentation ions, and comparing the obtained data with commercial standards and available literature. The data obtained were analyzed using the Waters MassLynx v.4.1 software (Waters, Milford, MA, USA). All analyzes were performed in triplicate (n = 9). The MS spectrophotometer operated under negative ionization conditions, and the analysis was performed by scanning the full spectrum within the range (*m/z*) from 120 to 2000. The quantitative analysis was based on a standard curve for ferulic acid, caffeic acid, and vitexin as standard substances.

### 3.6. Colour Measurement in the CIE L*a*b* Space Analysis and Color Differences (ΔΕ)

The instrumental color analysis was performed based on the CIE L*a*b* system using the colorimeter UltraScan VIS HunterLAB, according to the standards CIE 15:2004 and ISO 7724/1. Analyses were carried out in the reflected light within 400–700 nm (slot 25 mm, diffuse reflection coefficient 8°). The color difference (parameter ΔΕ) was calculated from the formula: ΔΕ = √((ΔL)^2 + (Δa)^2 + (Δb)^2).

### 3.7. Statistical Analysis

Data were expressed as mean ± standard deviations (n = 9). The results obtained were analyzed using Statistica 13.3. software (TIBCO Software Inc., Palo Alto, CA, USA) by a three-way analysis of variance (ANOVA) based on Duncan’s post-hoc test at a significance level of *p* < 0.05.

## 4. Conclusions

The study showed that celeriac peeling, the time the pulp was kept at 25 °C and the pectinase treatment had a significant effect on the pressing yield, total polyphenol content, the level of phenolic compounds, and the antioxidant potential of the celeriac juices obtained. The application of the enzyme in the unpeeled raw material pulp resulted in a slight increase in juice yield and a visible decrease in the peeled root pulp compared to the control. After enzymatic maceration of the peeled and unpeeled celeriac pulp, there was twice the antioxidant activity of DPPH^*^, a lower content of total polyphenols, and, in the peeled celeriac juices, a higher ferric ion reduction ability (FRAP).

A total of 20 phenolic compounds were identified in all celeriac juices. The pectinase treatment of peeled and unpeeled celeriac pulp resulted in a significant increase in quinic acid and a decrease in flavonoid content. Enzyme treatment of peeled celeriac pulp had no significant effect on the levels of apigenins and luteolins, while in juices from unpeeled raw material, their content decreased compared to identical control samples. The colour parameters a* and b* were strongly correlated with the juices’ total apigenin and luteolin content. The best quality characteristics of the juices, including significantly higher antioxidant potential and phenolic compound content, were obtained from unpeeled celery pulp incubated at 25 °C for 30 min, including pectinase.

The application of the enzyme to the celeriac pulp for a short 30-min period and a low incubation temperature of 25 °C produced a juice characterized by a high content of health-promoting and thermolabile compounds. In addition, a higher pressing efficiency of the pulp juice was achieved. This creates the potential to bring innovative vegetable juices to the market. With that said, further research into modeling the organoleptic characteristics of juices using enzymes, including combining the action of pectinase with other enzymes, is desirable.

## Figures and Tables

**Table 1 molecules-27-08610-t001:** Effect of pectinase on celeriac pulp on pressing yield, pH, content of total soluble solids, and selected sugars.

Sample Name	Delivery Capacity [%]	pH	Extract [°Bx]	Fructose [g/L]	Glucose [g/L]	Sucrose [g/L]
PC	47 ± 0.5 ^a^	5.9 ± 0.0 ^c^	6.9± 0.1 ^a^	3.1± 0.3 ^ab^	16.0± 0.0 ^b^	20.6 ± 0.6 ^cd^
PC1	64 ± 0.3 ^d^	5.8 ± 0.2 ^c^	6.7 ± 0.2 ^a^	3.1 ± 0.2 ^ab^	16.5 ± 0.1 ^bc^	20.9 ± 0.5 ^d^
PC2	67± 0.1 ^de^	5.8 ± 0.2 ^c^	6.9 ± 0.1 ^a^	3.3 ± 0.2 ^b^	17.5 ± 0.1 ^c^	22.1 ± 0.1 ^e^
UC	49 ± 0.2 ^ab^	6.0 ± 0.1 ^c^	8.3 ± 0.3 ^f^	5.2± 0.5 ^c^	22.1 ± 0.3 ^d^	19.3 ± 0.2 ^bc^
UC1	57 ± 0.1 ^c^	5.9 ± 0.6 ^c^	8.0 ± 0.2 ^e^	5.7 ± 0.3 ^cd^	25.0 ± 0.0 ^e^	20.9 ± 0.5 ^d^
UC2	60 ± 0.0 ^c^	5.9 ± 0.5 ^c^	8.1 ± 0.1 ^ef^	6.3 ± 0.3 ^d^	23.6 ± 0.7 ^de^	15.7 ± 0.4 ^a^
PCE1	47 ± 0.1 ^a^	4.2 ± 0.2 ^b^	7.2 ± 0.2 ^b^	2.8 ± 0.1 ^a^	14.9 ± 0.4 ^a^	20.2 ± 0.1 ^cd^
PCE2	53 ± 0.4 ^b^	4.2 ± 0.3 ^b^	7.4 ± 0.3 ^c^	2.8 ± 0.8 ^a^	15.9 ± 0.4 ^b^	22.0 ± 0.9 ^e^
UCE1	59 ± 0.6 ^c^	3.9 ± 0.0 ^a^	8.3 ± 0.4 ^f^	5.6 ± 0.8 ^cd^	24.8 ± 0.0 ^e^	20.2 ± 0.1 ^cd^
UCE2	70± 0.3 ^e^	3.9 ± 0.1 ^a^	7.7 ± 0.1 ^d^	5.7 ± 0.2 ^cd^	22.8 ± 0.2 ^d^	18.8 ± 0.0 ^b^
Three-factor analysis of variance ANOVA						
Factor 1	0.0000	0.0000	0.0001	0.0000	0.0000	0.0162
Factor 2	0.0003	0.0027	0.0000	0.0000	0.0000	0.0000
Factor 3	0.0000	1.0000	0.6944	0.0000	0.0000	0.0000
Factor 1 × Factor 2	0.0000	0.0000	0.0000	0.6043	0.0000	0.0000
Factor 1 × Factor 3	0.0039	1.0000	0.0128	0.0000	0.0000	0.0000
Factor 2 × Factor 3	0.1453	1.0000	0.0001	0.0000	0.0000	0.0000
Factor 1 × Factor 2 × Factor 3	0.1453	1.0000	0.0004	0.0089	0.0000	0.0000

Each value is expressed as mean ± SD (n = 9). Means in the same column with no common superscript differed significantly (*p* < 0.05); Factor 1—enzyme; Factor 2—Type of celeriac; Factor 3—Time; Factor 1 × Factor 2—Interaction between enzyme and time; Factor 1 × Factor 3—Interaction between enzyme and type of celeriac; Factor 2 × Factor 3—Interaction between celeriac type and time; Factor 1 × Factor 2 × Factor 3—Interactions between enzyme, celeriac type and time. Abbreviation: PC—juice pressed from peeled celery root, no enzyme, zero incubation time, without incubation in the incubator; PC1—juice pressed from peeled celery root, no enzyme, 30 min incubation in an incubator, incubation temperature of 25 °C; PC2—juice pressed from peeled celery root, no enzyme, 60 min incubation in an incubator, incubation temperature of 25 °C; UC—juice pressed from unpeeled celery root, no enzyme, zero incubation time, without incubation in the incubator; UC1—juice pressed from unpeeled celery root, no enzyme, 30 min incubation in an incubator, incubation temperature of 25 °C; UC2—juice pressed from unpeeled celery root, no enzyme, 60 min incubation in an incubator, incubation temperature of 25 °C; PCE1—juice pressed from peeled celery root, enzyme pectinase (dose: 10 mg/100 g pulp), 30 min incubation in an incubator, incubation temperature of 25 °C; PCE2—juice pressed from peeled celery root, enzyme pectinase (dose: 10 mg/100 g pulp), 60 min incubation in an incubator, incubation temperature of 25 °C; UCE1—juice pressed from unpeeled celery root, enzyme pectinase (dose: 10 mg/100 g pulp), 30 min incubation in an incubator, incubation temperature of 25 °C; UCE2—juice pressed from unpeeled celery root, enzyme pectinase (dose: 10 mg/100 g pulp), 60 min incubation in an incubator, incubation temperature of 25 °C.

**Table 2 molecules-27-08610-t002:** Effect of pectinase on celeriac pulp on antioxidant properties and total polyphenol content.

Sample Name	ABTS^*+^ [µmol Trolox/L]	DPPH^*^ [µmol Trolox/L]	FRAP [µmol Trolox/L]	Total Polyphenol [mg GAE/L]
PC	4759 ± 18 ^e^	578 ± 8 ^a^	101 ± 1 ^a^	147 ± 7 ^de^
PC1	4041 ± 17 ^ab^	476 ± 23 ^a^	97 ± 3 ^a^	131 ± 6 ^d^
PC2	4505 ± 8 ^cd^	656 ± 2 ^a^	96 ± 2 ^a^	126 ± 6 ^cd^
UC	4638 ± 12 ^de^	566 ± 9 ^a^	401 ± 2 ^f^	225 ± 1 ^f^
UC1	4137 ± 17 ^abc^	551 ± 9 ^a^	327 ± 7 ^e^	177 ± 8 ^e^
UC2	3944 ± 16 ^ab^	449 ± 6 ^a^	193± 2 ^d^	164 ± 3 ^e^
PCE1	3666 ± 37 ^a^	1108 ± 4 ^b^	125 ± 2 ^b^	112 ± 1 ^bc^
PCE2	4626 ± 21 ^d^	950 ± 18 ^b^	123 ± 2 ^b^	105 ± 2 ^ab^
UCE1	4367 ± 40 ^bcd^	938 ± 16 ^b^	157 ± 5 ^c^	117 ± 6 ^bcd^
UCE2	3847 ± 25 ^a^	587 ± 12 ^a^	193 ± 2 ^d^	93 ± 5 ^a^
Three-factor analysis of variance ANOVA				
Factor 1	0.7693	0.0000	0.0000	0.0000
Factor 2	0.1982	0.0066	0.0000	0.0000
Factor 3	0.0975	0.06021	0.0000	0.0033
Factor 1 × Factor 2	0.3539	0.0782	0.0000	0.0000
Factor 1 × Factor 3	0.6817	0.0142	0.0000	0.3623
Factor 2 × Factor 3	0.0001	0.0413	0.0000	0.1037
Factor 1 × Factor 2 × Factor 3	0.0595	0.6792	0.0000	0.6324

Each value is expressed as mean ± SD (n = 9). Means in the same column with no common superscript differed significantly (*p* < 0.05); Factory as above.

**Table 3 molecules-27-08610-t003:** Identification of phenolic compounds in celeriac juice.

Identification	t_r_ (min)	[M-H]^−^ (*m/z*)	[M-H]^-^ MS/MS (*m/z*)	Λ_max_ (nm)
Caffeic acid hexoside	0.78	341	179,135	323
Quinic acid	1.01	191	163	nd
Coumaroylquinic acid	1.83	337	191,163	228,312
Dicaffeic acid	3.05	341	191	286,338
Ferulic acid	3.11	193	178,149	281,324
Caffeoylsinpylquinic acid	4.75	559	341,179	275,330
Chrysoeriol	2.81	299	257,169	281,308
Apigenin-6-C-glucoside	3.91	431	341,311,269	335
Hydroxydimethoxycoumarin	4.13	221	206,261	290,336
Kaempferol-3,7-O-diglucoside	4.32	611	151,285	267,290
Luteolin-7-O-glucoside	4.99	447	285	274,348
Chrysoeriol-7-O-6′′-malonyl glucoside	5.14	547	299	284,322
Apiin	5.47	563	431,269,225	266,334
Taxifolin hexoside I	5.65	465	447,303,285,259	322
Chrysoeriol-7-O-apiosylglucoside	5.85	593	285	265,328,346
Acetylated apigenin-C-hexoside-O-pentoside	6.00	605	545,431,311,269	324,338
Acetylated luteolin hexoxyl-rhamnoside	6.32	635	299,284	338,345
Taxifolin hexoside II	6.62	465	285,303,447	267,276,321,338
Luteolin-7-O-malonyl-apiosylglucoside b	6.91	665	285	267,276,283
Chrysoeriol-7-O-glucoside	7.06	461	299	267,276,284

**Table 4 molecules-27-08610-t004:** Effect of pectinase on celeriac pulp on the content of selected phenolic acids, apigenins, and luteolins in juice.

Sample Name	Quinic Acid [mg/L]	Dicaffeic Acid [mg/L]	Ferulic Acid [mg/L]	Total Phenolic Acids [mg/L]	Apigenin-6-C-glucoside [mg/L]	Apiin [mg/L]	Acetylated apigenin-C-hexoside-O-pentoside [mg/L]	Total Apigenins [mg/L]	Luteolin-7-O-glucoside [mg/L]	Acetylated Luteolin hexoxyl-rhamnosyde [mg/L]	Luteolin-7-O-malonyl-apiosyl glucoside b [mg/L]	Total Luteolins [mg/L]
PC	47.9 ± 0.4 ^a^	6.0± 0.1 ^d^	5.2 ± 0.2 ^c^	59.1 ± 0.2 ^ab^	0.8 ± 0.1 ^cd^	3.3 ± 0.1 ^b^	0.3 ± 0.1 ^a^	4.4 ± 0.2 ^bc^	0.5 ± 0.1 ^a^	0.2± 0.1 ^a^	0.3 ± 0.2 ^ab^	1.0 ± 0.2 ^ab^
PC1	45.7± 0.7 ^a^	1.2 ± 0.0 ^c^	14.1 ± 0.2 ^e^	61.0 ± 0.1 ^b^	0.6 ± 0.2 ^c^	3.8 ± 0.3 ^c^	0.4 ± 0.0 ^ab^	4.8 ± 0.2 ^c^	0.6± 0.2 ^a^	0.1 ± 0.3 ^a^	0.1 ± 0.2 ^a^	0.8 ± 0.2 ^a^
PC2	43.7± 0.2 ^a^	0.1 ± 0.0 ^a^	11.5 ± 0.8 ^e^	55.3 ± 0.7 ^a^	0.2 ± 0.1 ^a^	3.6 ± 0.1 ^bc^	0.4 ± 0.2 ^ab^	4.2 ± 0.3 ^b^	0.6 ± 0.1 ^a^	0.1 ± 0.0 ^a^	0.1 ± 0.3 ^a^	0.8 ± 0.1 ^a^
UC	106.5 ± 0.3 ^c^	0.2± 0.0 ^ab^	4.1 ± 0.4 ^b^	110.8 ± 0.9 ^d^	1.2 ± 0.2 ^d^	0.4 ± 0.0 ^a^	0.4 ± 0.2 ^ab^	2.0 ± 0.4 ^a^	4.8 ± 0.1 ^d^	0.5 ± 0.0 ^b^	0.5 ± 0.1 ^b^	5.8 ± 0.5 ^d^
UC1	109.1 ± 0.9 ^cd^	0.0 ± 0.0 ^a^	4.4 ± 0.2 ^b^	113.5 ± 0.3 ^de^	0.9 ± 0.2 ^cd^	5.1 ± 0.0 ^e^	0.3 ± 0.01 ^a^	6.3 ± 0.2 ^d^	3.0 ± 0.2 ^c^	0.4 ± 0.0 ^b^	0.3 ± 0.1 ^ab^	3.7 ± 0.3 ^c^
UC2	84.9± 0.3 ^b^	0.2 ± 0.0 ^ab^	4.2 ± 0.0 ^b^	89.3 ± 0.2 ^c^	1.2 ± 0.3 ^d^	6.0 ± 0.0 ^f^	0.4 ± 0.0 ^ab^	7.6 ± 0.5 ^e^	2.7 ± 0.3 ^c^	0.2 ± 0.0 ^a^	0.2 ± 0.0 ^ab^	3.1 ± 0.0 ^c^
PCE1	132.4± 0.9 ^e^	0.1 ± 0.0 ^a^	9.6 ± 0.0 ^d^	142.1± 0.9 ^f^	0.9 ± 0.2 ^cd^	3.1 ± 0.2 ^b^	0.2 ± 0.0 ^a^	4.2 ± 0.1 ^b^	0.5 ± 0.2 ^a^	0.3 ± 0.2 ^ab^	0.1 ± 0.0 ^a^	0.9 ± 0.0 ^a^
PCE2	100.6 ± 0.1 ^c^	0.1 ± 0.0 ^a^	9.4 ± 0.2 ^d^	110.1 ± 0.5 ^d^	0.3 ± 0.0 ^b^	4.6 ± 0.3 ^d^	0.2 ± 0.0 ^a^	5.1 ± 0.2 ^c^	0.5 ± 0.1 ^a^	0.1 ± 0.0 ^a^	0.0 ± 0.2 ^a^	0.6± 0.2 ^a^
UCE1	139.5 ± 0.2 ^e^	0.1 ± 0.0 ^a^	2.5 ± 0.0 ^a^	142.1 ± 0.3 ^f^	0.7 ± 0.1 ^c^	4.9 ± 0.8 ^de^	0.2± 0.0 ^a^	5.8 ± 0.4 ^d^	0.5 ± 0.1 ^a^	0.1 ± 0.0 ^a^	0.1 ± 0.0 ^a^	0.7 ± 0.0 ^a^
UCE2	118.5 ± 0.1 ^d^	0.1 ± 0.0 ^a^	2.2 ± 0.2 ^a^	120.8 ± 0.3 ^e^	0.8± 0.1 ^cd^	3.9 ± 0.4 ^c^	0.3± 0.2 ^a^	5.0 ± 0.2 ^c^	1.1 ± 0.1 ^b^	0.1 ± 0.0 ^a^	0.1 ± 0.0 ^a^	1.2 ± 0.2 ^b^
Three-factor analysis of variance ANOVA												
Factor 1	0.0000	0.0000	0.0006	0.0000	0.3447	0.0000	0.0000	0.0000	0.0000	0.0000	0.0000	0.0000
Factor 2	0.0000	0.0000	0.0000	0.0000	0.0000	0.0000	0.0065	0.0000	0.0000	0.0000	0.0000	0.0000
Factor 3	0.0000	0.0000	0.0000	0.0000	0.0004	0.0000	0.0002	0.0039	0.1468	0.0000	0.0000	0.0154
Factor 1 × Factor 2	0.0000	0.0000	0.0024	0.0000	0.0000	0.0000	0.0140	0.0000	0.0000	0.0000	0.0000	0.0000
Factor 1 × Factor 3	0.0057	0.0000	0.0000	0.0018	0.0058	0.3221	0.2102	0.0293	0.0001	0.0019	0.0003	0.0000
Factor 2 × Factor 3	0.4116	0.0000	0.0000	0.2166	0.0000	0.0000	0.0004	0.9536	0.0476	0.4211	0.0000	0.0967
Factor 1 × Factor 2 × Factor 3	0.0011	0.0000	0.0000	0.0005	0.5951	0.0000	0.0005	0.0000	0.0001	0.0000	0.0000	0.0000

Each value is expressed as mean ± SD (n = 9). Means in the same column with no common superscript differed significantly (*p* < 0.05); Factory as above.

**Table 5 molecules-27-08610-t005:** Effect of pectinase on celeriac pulp on CIE L*a*b* color parameters and differences (ΔΕ*) in juice.

Sample Name	L*	a*	b*	ΔΕ*
PC	33.44 ± 0.02 ^ef^	−0.01 ± 0.03 ^d^	4.94 ± 0.04 ^e^	-
PC1	33.60 ± 0.04 ^f^	−0.12 ± 0.13 ^d^	5.51 ± 0.07 ^f^	0.60
PC2	33.11 ± 0.10 ^de^	−0.62 ± 0.02 ^c^	4.63 ± 0.07 ^d^	0.76
UC	36.95 ± 0.02 ^h^	2.22 ± 0.02 ^g^	9.54 ± 0.04 ^i^	-
UC1	33.28 ± 0.33 ^e^	0.96 ± 0.06 ^f^	7.48 ± 0.22 ^g^	4.39
UC2	34.62 ± 0.02 ^g^	0.63 ± 0.02 ^e^	7.65 ± 0.02 ^h^	3.40
PCE1	32.76 ± 0.06 ^c^	−0.93 ± 0.01 ^a^	3.17 ± 0.01 ^b^	2.11
PCE2	32.89 ± 0.03 ^cd^	−0.96 ± 0.02 ^a^	2.82 ± 0.02 ^a^	2.38
UCE1	32.19 ± 0.10 ^b^	−0.78 ± 0.00 ^b^	4.65 ± 0.04 ^d^	7.45
UCE2	31.87 ± 0.15 ^a^	−0.81 ± 0.01 ^b^	4.44 ± 0.04 ^c^	7.81
Three-factor analysis of variance ANOVA				
Factor 1	0.0000	0.0000	0.0000	-
Factor 2	0.1049	0.0000	0.0000	-
Factor 3	0.0110	0.0000	0.0000	-
Factor 1 × Factor 2	0.0000	0.0000	0.0000	-
Factor 1 × Factor 3	0.0003	0.0000	0.3355	-
Factor 2 × Factor 3	0.0000	0.0509	0.0000	-
Factor 1 × Factor 2 × Factor 3	0.0000	0.0680	0.0000	-

Each value is expressed as mean ± SD (n = 9). Means in the same column with no common superscript differed significantly (*p* < 0.05); Factory as above.

## Data Availability

The data are included in the manuscript.

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
