# Peer review of "Effect of Celeriac Pulp Maceration by Rhizopus sp. Pectinase on Juice Quality"

_molecules, 2022, doi:10.3390/molecules27238610_

Round 1

Reviewer 1 Report

The work submitted for review is interesting and well written. Actually, there are no serious charges.

I only miss the emphasis on the practical purpose of such research. For the aim of the work, the authors should present a general, practical purpose in addition to the experimental purpose. At the same time, the conclusions should end with such a general practical summary.

Author Response

Dear Sirs,

thank you very much for sending your review. We have complied with the comments and are sending the individual responses and the revised document for review.

Manuscript ID: molecules-2070513

Title: Effect of celeriac pulp maceration by Rhizopus sp. pectinase on juice quality.

Authors: Grażyna Jaworska, Natalia Szarek*, Paweł Hanus

Received: 16 November 2022

E-mails: gjaworska@ur.edu.pl, nataliasz@dokt.ur.edu.pl, phanus@ur.edu.pl

Submitted to section: Natural Products Chemistry,

https://www.mdpi.com/journal/molecules/sections/natural_products_chemistry

Bioactives and Functional Ingredients in Foods II

https://www.mdpi.com/journal/molecules/special_issues/ingredients_food_II

Response to review 1.

„I only miss the emphasis on the practical purpose of such research. For the aim of the work, the authors should present a general, practical purpose in addition to the experimental purpose. At the same time, the conclusions should end with such a general practical summary.”

Add to the conclusions (line 162).

The application of the enzyme to the celery pulp for a short 30-minute period and a low incubation temperature of 25ËšC produced a juice characterised by a high content of health-promoting and thermolabile compounds. In addition, a higher pressing efficiency of the pulp juice was achieved. This creates the potential to bring innovative vegetable juices to the market. With that said, further research into modelling the organoleptic characteristics of juices using enzymes, including combining the action of pectinase with other enzymes, is desirable.

Thank you very much,
Yours faithfully,
Natalia Szarek

Reviewer 2 Report

You must carry out several corrections, particularly in the format. 

Remember that the name of the microorganisms must be in italics. 

Add the explanation of all the abbreviations in the tables.

Can you explain the means of "All samples" in table 3.

In conclusion section, avoid repeating the results.

Author Response

Dear Sirs,

thank you very much for sending your review. We have complied with the comments and are sending the individual responses and the revised document for review.

Manuscript ID: molecules-2070513

Title: Effect of celeriac pulp maceration by Rhizopus sp. pectinase on juice quality.

Authors: Grażyna Jaworska, Natalia Szarek*, Paweł Hanus

Received: 16 November 2022

E-mails: gjaworska@ur.edu.pl, nataliasz@dokt.ur.edu.pl, phanus@ur.edu.pl

Submitted to section: Natural Products Chemistry,

https://www.mdpi.com/journal/molecules/sections/natural_products_chemistry

Bioactives and Functional Ingredients in Foods II

https://www.mdpi.com/journal/molecules/special_issues/ingredients_food_II

Response to review 2.

„Remember that the name of the microorganisms must be in italics. 

Add the explanation of all the abbreviations in the tables.

Can you explain the means of "All samples" in table 3.

In conclusion section, avoid repeating the results.”

Line 14-15: Peeling, enzyme addition and maceration time significantly affected the quality characteristics of the juice. 

corrected to: Peeling, enzyme addition, and maceration time significantly affected the quality characteristics of the juice. 

Line 15-16: The juice obtained from peeled celeriac was characterised by higher 15 pressing yield, sucrose content, antioxidant activity (ABTS*+ and DPPH*), 

corrected to: The juice obtained from peeled celeriac was characterised by higher pressing yield, sucrose content, antioxidant activity (ABTS*+ and DPPH*),  

* changes throughout the paper.

Line 91: Aspergillus sp. Pectinase 

Corrected to: Aspergillus sp. pectinase 

Line 92: Aspergillus sp. pectinase has the highest activity at pH 4.0 and 50°C. 

Corrected to: Aspergillus sp. pectinase has the highest activity at pH 4.0 and 50°C. 

Line 95. Rhizopus sp. pectinase is used for enzymatic digestion of the cell 95 wall. 

Corrected to: Rhizopus sp. pectinase is used for enzymatic digestion of the cell 95 wall. 

Table 1 (and table number 2, 4 and 5)

As suggested by the Reviewer, descriptions and explanations of abbreviations of sample names have been added to Table 1. As authors, we feel that the use of explanations of abbreviations under each table is unnecessary, and that the text will add considerably to its volume. If the Editor and Reviewer feel that this change is important, we, as authors, are able to accommodate the Reviewer's suggestion, and include appropriate descriptions under each table.

Add to the table 1 (Line 155). Abbreviation: PC- juice pressed from peeled celery root, no enzyme, zero incubation time, without incubation in the incubator; PC1- juice pressed from peeled celery root, no enzyme, 30 minutes incubation in an incubator, incubation temperature of 25°C; PC2- juice pressed from peeled celery root, no enzyme, 60 minutes incubation in an incubator, incubation temperature of 25°C; UC- juice pressed from unpeeled celery root, no enzyme, zero incubation time, without incubation in the incubator; UC1- juice pressed from unpeeled celery root, no enzyme, 30 minutes incubation in an incubator, incubation temperature of 25°C; UC2- juice pressed from unpeeled celery root, no enzyme, 60 minutes incubation in an incubator, incubation temperature of 25°C; PCE1- juice pressed from peeled celery root, enzyme pectinase (dose: 10 mg/100 g pulp), 30 minutes incubation in an incubator, incubation temperature of 25°C; PCE2- juice pressed from peeled celery root, enzyme pectinase (dose: 10 mg/100 g pulp), 60 minutes incubation in an incubator, incubation temperature of 25°C; UCE1- juice pressed from unpeeled celery root, enzyme pectinase (dose: 10 mg/100 g pulp), 30 minutes incubation in an incubator, incubation temperature of 25°C; UCE2- juice pressed from unpeeled celery root, enzyme pectinase (dose: 10 mg/100 g pulp), 60 minutes incubation in an incubator, incubation temperature of 25°C. 

* changed to italics throughout the paper

Line 205-216: Apart from ferulic acid (m/z 193), quinic acid (m/z 191) and dicaffeic acid (m/z 341), there were also: caffeic acid hexoside (m/z 341), coumaroylquinic acid (m/z 337) and caffeoylsinpylquinic acid (m/z 559). Celeriac juices contained also chrysoeriol (m/z 299), chrysoeriol-7-O-6’’-malonyl glucoside (m/z 547), chrysoeriol- 7-O-apiosylglucoside (m/z 593), and chrysoeriol-7-O-glucoside (m/z 461). Among luteolins, the following were identified: luteolin-7-O-malonyl-apiosylglucoside B (m/z 665), luteolin-7-O-glucoside (m/z 447), acetylated luteolin hexoxyl-rhamnoside (m/z 635), and luteolin-7-O-malonyl-apiosylglucoside B (m/z 635). The juices also included apigenins such as apigenin-6-C-glucoside (m/z 431), apiin (m/z 563), and acetylated apigenin-C-hexoside-O-pentoside (m/z 605). Furthermore, hydroxydimethoxycoumarin (m/z 221), kaempferol-3,7-O-diglucoside (m/z 611), and taxifolin hexoside I and II (m/z 465) were also identified in such juices. 

Corrected to: Apart from ferulic acid (m/z 193), quinic acid (m/z 191) and dicaffeic acid (m/z 341), there were also: caffeic acid hexoside (m/z 341), coumaroylquinic acid (m/z 337) and caffeoylsinpylquinic acid (m/z 559). Celeriac juices contained also chrysoeriol (m/z 299), chrysoeriol-7-O-6’’-malonyl glucoside (m/z 547), chrysoeriol- 7-O-apiosylglucoside (m/z 593), and chrysoeriol-7-O-glucoside (m/z 461). Among luteolins, the following were identified: luteolin-7-O-malonyl-apiosylglucoside B (m/z 665), luteolin-7-O-glucoside (m/z 447), acetylated luteolin hexoxyl-rhamnoside (m/z 635), and luteolin-7-O-malonyl-apiosylglucoside B (m/z 635). The juices also included apigenins such as apigenin-6-C-glucoside (m/z 431), apiin (m/z 563), and acetylated apigenin-C-hexoside-O-pentoside (m/z 605). Furthermore, hydroxydimethoxycoumarin (m/z 221), kaempferol-3,7-O-diglucoside (m/z 611), and taxifolin hexoside I and II (m/z 465) were also identified in such juices. 

Change in table 3: [M-H]- m/z  

Corrected to: [M-H]- m/z 

Change in table 3: [M-H]- MS/MS (m/z) 

Corrected to: [M-H]- MS/MS (m/z

Page number 12, Line 35: Rhizopus sp. were supplied by... 

Corrected to: Rhizopus sp. were supplied by... 

Table 3. “All samples” in table 3.

In accordance with the Reviewer's suggestion, we have decided to delete the "Sample name" column in Table 3. On the other hand, in the text of the article, in line 205, we propose to insert the following sentence: 'Each phenolic compound listed in Table 3 was identified in all juices analysed pressed from both peeled and unpeeled celery'.

 Conclusions:

In line with the Reviewer's comment, the summary has been revised avoiding repetition of results from the summary section. A new summary is proposed below:

The study showed that celeriac peeling, the time the pulp was kept at 25°C and the pectinase treatment had a significant effect on the pressing yield, total polyphenol content, the level of phenolic compounds, and the antioxidant potential of the celeriac juices ob-tained. The application of the enzyme in the unpeeled raw material pulp resulted in a slight increase in juice yield, and a visible descrease in the peeled root pulp compared to the control. After enzymatic maceration of the peeled and unpeeled celeriac pulp, there was twice the antioxidant activity of DPPH*, a lower content of total polyphenols and, in the peeled celeriac juices, a higher ferric ion reduction ability (FRAP).

A total of 20 phenolic compounds were identified in all celeriac juices. The pectinase treatment of peeled and unpeeled celeriac pulp resulted in a significant increase in quinic acid and a decrease in flavonoid content. Enzyme treatment of peeled celeriac pulp had no significant effect on the levels of apigenins and luteolins, while in juices from unpeeled raw material, their content decreased compared to identical control samples. The colour parameters a* and b* were strongly correlated with the juices' total apigenin and luteolin content. The best quality characteristics of the juices, including significantly higher anti-oxidant potential and phenolic compound content, were obtained from unpeeled celery pulp incubated at 25ËšC for 30 minutes, including pectinase.

Thank you very much,
Yours faithfully,
Natalia Szarek

Reviewer 3 Report

The report shows novel and interesting results on the use of pectinase form Rhizopus sp. for improving yield and quality of the juice from celeriac pulp.

Overall, the manuscript was well prepared with very few writing mistakes. The experiments were well designed with the uses of reliable analytical methods. The results are also well presented and sufficiently discussed.

However, some minor revisions are suggested for improving the quality of the paper as shown below:

1.      The sentence from line 20-23 in the abstract is confusing and should be rewritten.

2.      A concluding sentence should be provided for the Abstract.

3.      The first paragraph from line 81-91 of the Results and Discussion should be removed or concisely rewritten and move to the Introduction.

4.      The presentation of ANOVA analyses in the Results and Discussion is not necessary. These should be provided as Supplementary data.

5.      The repetitions of the information of the spectrophotometer in section 3.5.2, 3.5.3 and 3.5.4 are not necessary. The term “above spectrophotometer” or “the same spectrophotometer in section 3.5.1” is more reasonable.

6.      A concluding sentence emphasizing the significance, importance and potential applications of the findings should be provided in the Conclusion section. Some recommendations for further research may also be added to the end of the Conclusion section.

Author Response

Dear Sirs,

thank you very much for sending your review. We have complied with the comments and are sending the individual responses and the revised document for review.

Manuscript ID: molecules-2070513

Title: Effect of celeriac pulp maceration by Rhizopus sp. pectinase on juice quality.

Authors: Grażyna Jaworska, Natalia Szarek*, Paweł Hanus

Received: 16 November 2022

E-mails: gjaworska@ur.edu.pl, nataliasz@dokt.ur.edu.pl, phanus@ur.edu.pl

Submitted to section: Natural Products Chemistry,

https://www.mdpi.com/journal/molecules/sections/natural_products_chemistry

Bioactives and Functional Ingredients in Foods II

https://www.mdpi.com/journal/molecules/special_issues/ingredients_food_II

Response to review 3.

  1. „The sentence from line 20-23 in the abstract is confusing and should be rewritten”.

 Changed in abstract.

Line 20-23: Maceration of the pulp with pectinase, compared to control samples held at 25°C for the same time, increased the pressing efficiency by 2-10% in juices from unpeeled celeriac, the antioxidant potential (FRAP) by 22% in juices from the peeled root, by the DPPH* method by 24-57% in all juices tested, and the total phenolic acid content by 20-57%.  

Corrected to: 

The addition of pectinase to unpeeled celery pulp resulted in a 2-10% increase in pressing efficiency compared to the control sample held at 25ËšC for the same period of time. Maceration of the enzyme-peeled pulp increased the antioxidant potential of the juice by 22% in the FRAP method. In contrast, in all juices analysed, unpeeled and peeled roots increased antioxidant activity measured by the DPPH* method by 24-57% and total phenolic acids by 20-57%.

2.  „A concluding sentence should be provided for the Abstract”.

 Add a concluding sentence in the abstract (line 26).

Short-term, 30-minute maceration of peeled and unpeeled celery pulp with pectinase from Rhizopus sp. had a significant effect on increasing juice yield, antioxidant activity and phenolic compound content. 

3. „The first paragraph from line 81-91 of the Results and Discussion should be removed or concisely rewritten and move to the Introduction”.

Rewrite the paragraph from the results and discussion to the introduction.

Line 81- 91: Pectinolytic enzymes were already used to produce fruit juices in the 1930s. They were used most often to depectinize pome fruit juice and to process pulp and berry juice [26]. A properly selected enzyme preparation or enzyme supports the pressing process, prevents the clouding of juices caused by an excessive amount of hemicellulose, and facilitates clarification and filtration [16]. Pectinase treatment of apple and berry juices, as well as guava puree, significantly improved antioxidant capacity, total phenolic content, increased process efficiency, and sugar concentration, making the colour of the end products more attractive [27]. For vegetable tissue processing, liquid pectinolytic preparations are most often used. These are: Pectinex Ultra SP-L, Ultrazym AFP-L, Rapidase® Carrot, Rohament PL® , Panzym SMASH XXL® or ROHAPECT® MA Plus, as well as powdered Aspergillus sp. pectinase [25]. 

Corrected to: Change the line: 74.

4. “The presentation of ANOVA analyses in the Results and Discussion is not necessary. These should be provided as Supplementary data”.

In the authors' opinion, the section on statistical elaboration pointed out by the Reviewer should remain in tables. The paper compares a number of factors, and putting this data in the tables allows the recipient to fully understand the results obtained. If the editor or the Reviewer feels that this part of the statistical elaboration should be moved to "supplementary data", we are able to agree with the suggestion, no less, in the authors' opinion, this statistical elaboration in the placed place is important for the understanding of the whole text of the article.

5. “The repetitions of the information of the spectrophotometer in section 3.5.2, 3.5.3 and 3.5.4 are not necessary. The term “above spectrophotometer” or “the same spectrophotometer in section 3.5.1” is more reasonable”.

In section 3.5.2, 3.5.3 and 3.5.4 remove repetitions: The absorbance was measured using the UV-1900 UV-Vis Shimadzu (Japan) spectrophotometer 

Corrected to: The absorbance was measured using the same spectrophotometer in section 3.5.1. 

6. „A concluding sentence emphasising the significance, importance and potential applications of the findings should be provided in the Conclusion section. Some recommendations for further research may also be added to the end of the Conclusion section”.

The application of the enzyme to the celery pulp for a short 30-minute period and a low incubation temperature of 25ËšC produced a juice characterised by a high content of health-promoting and thermolabile compounds. In addition, a higher pressing efficiency of the pulp juice was achieved. This creates the potential to bring innovative vegetable juices to the market. With that said, further research into modelling the organoleptic characteristics of juices using enzymes, including combining the action of pectinase with other enzymes, is desirable.

Thank you very much,
Yours faithfully,
Natalia Szarek
